# Intracoronary Thrombogenicity in Patients with Vasospastic Angina: An Observation Using Coronary Angioscopy

**DOI:** 10.3390/diagnostics11091632

**Published:** 2021-09-07

**Authors:** Hiroki Teragawa, Yuichi Orita, Chikage Oshita, Yuko Uchimura

**Affiliations:** Department of Cardiovascular Medicine, JR Hiroshima Hospital, 3-1-36 Futabanosato, Higashi-ku, Hiroshima 732-0057, Japan; orichyan@outlook.com (Y.O.); chikagekihara@gmail.com (C.O.); ymakita@hotmail.com (Y.U.)

**Keywords:** coronary spasm, angioscopy, coronary erosion, vasospastic angina

## Abstract

Background: Despite significant interest in intracoronary thrombi in patients with vasospastic angina (VSA), the phenomenon remains unclarified. Therefore, we investigated a possible relationship using coronary angioscopy (CAS) in VSA patients. Methods: Sixty patients with VSA, for whom we could assess the spastic segment using CAS, were retrospectively studied. An intracoronary thrombus on CAS was a white thrombus and an erosion-like red thrombus. We verified the clinical characteristics and lesional characteristics as they determined the risk of intracoronary thrombus formation. Results: There were 18 (30%) patients with intracoronary thrombi. More of the patients with intracoronary thrombi were male, current smokers and had severe concomitant symptoms; however, no statistically significant difference was observed upon logistic regression analysis. There were 18 (26%) coronary arteries with intracoronary thrombi out of 70 coronary arteries recognised in the spastic segments. Furthermore, atherosclerotic changes and segmental spasms were significant factors responsible for such lesions. Conclusion: Intracoronary thrombi occurred in 30% of VSA patients and much attention should be paid to the intracoronary thrombogenicity of VSA patients.

## 1. Introduction

Coronary spasm is characterised by transient vasoconstriction of the epicardial coronary artery, leading to myocardial ischaemia [1,2,3]. Coronary spasm plays a causal role in rest angina and exertional angina or acute coronary syndrome (ACS) [3]. Recently, much attention has focused on ischaemia with non-obstructive coronary arteries (INOCA) or myocardial infarction with non-obstructive coronary arteries (MINOCA) [4]. Coronary spasm is a crucial pathogenetic mechanism of INOCA and/or MINOCA [5,6,7,8,9]. Regarding the relationship between coronary spasm and ACS or MINOCA, much attention has been paid to the frequency of intracoronary thrombus formation in patients with vasospastic angina (VSA).

Many studies have investigated the relationship between intracoronary thrombus formation and coronary spasm using intracoronary imaging techniques that show intracoronary thrombi in patients with VSA [10,11,12,13,14]. The frequency of intracoronary thrombus formation in patients with VSA might vary depending on the studied patients’ characteristics, such as the presence of ACS or MINOCA or chronic coronary syndrome. Intracoronary imaging modalities include intravascular ultrasound (IVUS) [11], optical coherence tomography (OCT) [12,13,14] or coronary angioscopy (CAS) [10,14]. The assessment using IVUS or OCT could help determine the frequency of intracoronary thrombus formation in VSA patients, based on detailed observations of the entire circumference of the coronary artery and the whole of the coronary artery from the distal to proximal segments, especially in high-definition IVUS or OCT due to the high resolution of images. On the other hand, an assessment with CAS might help with direct visualisation of the vessel wall based on colour changes irrespective of the absence of an assessment of the whole coronary artery, especially in the distal segment of the artery.

In the present study, we investigated the frequency of intracoronary thrombus formation using CAS in VSA patients with coronary spasms among patients who underwent the routine spasm provocation test (SPT) and the clinical characteristics of such patients. Furthermore, we investigated the lesion characteristics with intracoronary thrombi at the spastic segments.

## 2. Material and Methods

### 2.1. Patient Selection

This retrospective, observational study occurred exclusively at our institution. We performed the SPT on 230 patients who had any chest symptoms from May 2016 to March 2020. The SPT aimed to clarify the origin of chest symptoms for appropriate medication, preventing sudden cardiac death. Among 230 patients who underwent the SPT, there were 164 patients with positive SPTs (Figure 1). Among 164 patients with positive SPTs, we excluded 92, including those who had undergone percutaneous coronary interventions (PCI, *n* = 4), had organic coronary stenosis (%stenosis ≥ 50%, *n* = 4), could not provide their informed consent (*n* = 24), had insufficient procedure time to perform the CAS study (*n* = 21) and for whom the operator judged the anatomy or run of the coronary artery was unsuitable for the CAS study (*n* = 39). There were four patients with MINOCA; however, we excluded these patients from the present study because of the safety of the CAS study. Thus, we assessed CAS in 72 patients. Among them, we could assess 60 patients with a spastic segment in at least one coronary vessel. We could not assess the other 12 patients within the spastic segment. However, we partially adopted the CAS data to investigate lesion characteristics with the non-spastic segment in the spastic vessel (defined as the SV-2). Our institution’s ethics committee approved the study protocol. All patients who underwent CAS and SPT provided written informed consent.

### 2.2. Coronary Angiography, SPT

An SPT was performed using the methods described previously [15]. In brief, vasodilator therapy was suspended at least 48 h before the SPT. After the initial coronary angiogram (CAG), 50, 100 and 200 μg doses of acetylcholine (ACh) were injected into the left coronary artery (LCA), then 20, 50 and 80 µg doses of ACh were injected into the right coronary artery (RCA). We obtained angiography just after coronary spasms were induced or the maximum ACh administration was finished. If a coronary spasm was induced but improved spontaneously, an intracoronary injection of 0.3 mg of nitroglycerin (NTG) was administered into the coronary artery before the final angiograms. If the coronary spasm provoked by ACh administration was prolonged or severe enough to induce hemodynamic instability, NTG was used to relieve the spasms since it was unavoidable. A positive SPT was a ≥90% narrowing of the coronary arteries on angiograms during provocation accompanied by usual chest pain and/or the presence of ST-segment deviation on ECG [3]. According to the positive SPT per vessel in the coronary artery, vessels with and without a positive SPT were defined as spastic vessels (SVs) and non-spastic vessels (NSVs), respectively. Furthermore, the SVs where the spastic segment was visible using CAS were defined as the SV-1. SV-2 referred to instances where the spastic segment was invisible using CAS.

The diameters of the coronary arteries were measured as described previously [15]. We selected spastic and atherosclerotic segments for quantitative analysis. In all cases, the luminal diameters were measured by a single investigator blinded to the clinical data, using an end-diastolic frame in a computer-assisted coronary angiographic analysis system (CAAS II/QUANTCOR; Siemens, Berlin, Germany). Lesions with >20% stenosis were defined as atherosclerotic lesions. In this study, low, moderate and high doses of ACh (L-ACh, M-ACh, H-ACh), were considered: 20, 50 and 80 μg for the RCA and 50, 100 and 200 μg for the LCA, respectively. A focal spasm was defined as a transient vessel narrowing of >90% limited to the major coronary arteries. A diffuse spasm was defined as a 90% diffuse vasoconstriction observed in ≥2 adjacent coronary segments of the coronary arteries [16]. In the present study, we also divided diffuse spasms into multiple focal spasms and diffuse spasms. The former was defined as the presence of multiple focal spasms in one coronary artery, and the latter was defined as the presence of pure diffuse spasms with similar degrees from proximal to distal segments. Thus, we also adopted another definition of spasm configuration: segmental, including focal and multiple focal spasms and diffuse spasms. Multi-vessel spasms were defined as coronary spasms that occurred in ≥2 major coronary arteries. We could not assess multi-vessel spasms when the subsequent SPT was negative after an unavoidable use of NTG. We also checked for subtotal and total occlusion due to coronary spasm and ST-T segment elevation on ECG.

### 2.3. Assessment of CAS

After administering NTG during the SPT, CAS examination was performed using a non-occluded type angioscope (Visible, FibreTech Co., Ltd., Tokyo, Japan) [14,17]. The outer section of the 4-Fr probing catheter (MEDIKIT Co., Ltd., Tokyo, Japan) was used as the guide to insert the optical fibre into the coronary artery. Angioscopic observations were made while the blood was cleared away from the view by injecting 3% dextran-40 through the probing catheter and guide catheter, as previously reported [17]. The optical fibre was placed at the distal segment of the coronary artery and slowly pulled back under angiographic guidance. CAS images were digitally recorded (Intertec Medicals Co. Ltd., Osaka, Japan). Plaques and thrombi were visualised under angiographic guidance and compared with the spasm site angiographically. Thrombi were defined based on the European Working Group on CAS criteria [18], and the presence of thrombi was assessed. Erosions were visually surrounded over the surface of vessel walls. However, because CAS could not assess the histopathology directly below such erosion, these erosion-like thrombi and white thrombi were assessed together as intracoronary thrombi (Figure 2). Maximum attention is required to eliminate the blood from the field of view and assess the presence of intracoronary thrombi. Patients were divided into two groups: Group I had no intracoronary thrombi and Group II had intracoronary thrombi. The plaque colour was graded as 0 (white), 1 (light yellow), 2 (yellow) and 3 (bright yellow) as previously described [19,20,21]. The maximum yellow grade of the plaques was assessed. Angioscopic evaluations were made by two angioscopic specialists who were blinded to the clinical status. If the judgements of the reviewers were discordant, the plaque colour was re-evaluated; however, if the re-evaluations remained discordant, the disagreement was resolved through discussions until a consensus was reached. We made efforts to perform the CAS observation, especially in the LAD, irrespective of the presence of coronary spasms; however, in the left circumflex coronary artery (LCX) and the RCA, due to the insufficient back up force of the guiding coronary artery, meandering coronary artery and small RCA, the CAS observation could not be performed sometimes.

### 2.4. Clinical Factors Assessed in the Present Study

The patient was asked about current smoking status, and any family history of coronary artery disease was recorded. Hypertension, dyslipidaemia, diabetes mellitus and chronic kidney disease were defined based on the standard definitions described in previous papers [22]. Blood chemical parameters were monitored routinely using blood drawn in the morning on CAG. Regarding the D-dimer value, the value < 0.5 µg/mL was adopted as the 0.5 µg/mL in the analyses. Coronary vasodilators, such as calcium-channel blockers, long-acting nitrate or nicorandil, statins and aspirin were assessed when the patient regularly visited the hospital before admission for CAG. The left ventricular ejection fraction was measured using cardiac ultrasonography. The number of angina attacks (per month) experienced in the preceding three months and the estimated duration (in months) from onset to the present admission were measured. Variant angina (VA), defined as angina with a recorded spontaneous ST elevation on ECG, and the severe concomitant symptoms of VSA, such as cold sweating or syncope, were also checked.

### 2.5. Statistical Analyses

Continuous data are presented as the mean ± SD or medians with interquartile ranges) for normally and non-normally distributed data. Baseline characteristics of the groups were compared using Student’s unpaired *t*-tests, Wilcoxon signed-rank tests or χ^2^ analysis, as appropriate. Multivariate logistic regression analyses of the presence of intracoronary thrombi were performed using factors with a *p*-value < 0.05 on the comparison between the two groups. Statistical analyses were performed using JMP Ver. 16 (SAS Institute Inc., Cary, NC, USA). *p*-values < 0.05 were considered statistically significant.

## 3. Results

Among the total 72 patients who underwent CAS study, one patient (1%) had a recurring coronary spasm during a CAS study because the proving catheter might stimulate the coronary artery at the myocardial bridge segment. The coronary spasm had been relieved by an intracoronary administration of NTG after a prompt observation of CAS.

Among 60 patients in whom the spastic segments could be assessed using CAS, 10 patients (17%) had white thrombi (Figure 2 left panel) and 12 (20%) had erosion-like red thrombi (Figure 2, centre and right panels). There were 18 patients (30%, Group II) with intracoronary thrombi. There were 42 patients without intracoronary thrombi on CAS in Group I.

### 3.1. Patients’ Characteristics in VSA Patients with Intracoronary Thrombi

Table 1 shows the clinical characteristics of Groups I and II. There were significantly more male participants and current smokers in Group II than in Group I (*p* < 0.01). Regarding the blood chemical parameters in the two groups, the serum levels of lipid-related parameters, glucose-related parameters, D-dimers, C-reactive protein and brain natriuretic peptide did not differ significantly between the two groups (Table 2). Regarding the medications before CAG, VSA-related symptoms and the presence of multi-vessel spasm (Table 3), the frequencies of vasodilator, statin or aspirin consumption did not differ significantly between the two groups. The median number of chest symptoms per month and the duration from the onset of chest symptoms to admission did not differ significantly between the two groups. The VA frequency did not differ significantly between the two groups; however, the frequency of severely concomitant symptoms was significantly higher in Group II than in Group I (*p* < 0.01). The frequency of multi-vessel spasms did not differ significantly between the two groups. There was no significant difference in the proportions of males (OR: 2.29, *p* = 0.13), current smokers (OR: 0.23, *p* = 0.63) and patients with concomitant symptoms (OR: 2.98, *p* = 0.08) between the two groups.

### 3.2. Lesion Characteristics with Intracoronary Thrombi

Among 72 patients, observation using CAS could be performed in 67 LAD, 8 LCX and 37 RCA, a total of 112 coronary vessels. According to the presence of coronary spasms and observation of CAS, there were 70 SV-1, 22 SV-2 and 20 NSV. The distribution of coronary artery vessels in which CAS could be performed differed in each of the three groups (Table 4). The frequencies of atherosclerotic change and yellow plaque and the degree of yellow plaque did not differ significantly between the three groups. However, intracoronary thrombi were detected only in SV-1 (*p* < 0.01, Table 4 and Figure 3). In addition, in 70 coronary arteries of 60 patients in whom CAS could be assessed within the spastic segment, there were 18 intracoronary thrombi (26%).

Regarding the only 70 SV-1, the frequency of yellow plaques and the degree of yellow plaques did not differ significantly between spastic segments with and without intracoronary thrombi. The atherosclerotic change frequency was significantly higher in the spastic segments with intracoronary thrombi than those without (*p* < 0.01, Table 5). Focal spasm and segmental spasm were also significantly higher in the spastic segments with intracoronary thrombi than those without intracoronary thrombi (both *p* < 0.01). The subtotal/total spasm frequency was higher in the spastic segments with intracoronary thrombi than those without intracoronary thrombi (*p* = 0.04). The ACh doses that could induce spasms tended to differ between the segments with and without intracoronary spasms (*p* = 0.07). The frequency of ST-T segment elevation on ECG during coronary spasm tended to be higher in the spastic segments with intracoronary thrombi than in those without (*p* = 0.08). Logistic regression analyses significantly associated segmental spasm (OR: 4.49, *p* = 0.03) and atherosclerotic change (OR: 4.11, *p* = 0.04) with spastic segments with intracoronary thrombi.

## 4. Discussion

To clarify the frequency of intracoronary thrombus formation in patients with VSA, we carried out investigations using CAS in patients with a positive SPT. The present study demonstrates that the frequency of intracoronary thrombus formation was 30% per patient and 26% per vessel in VSA patients under the condition that the spastic segments could be assessed using CAS. On comparisons between groups, we observed significantly more male participants, more participants who were current smokers and more participants with severe concomitant symptoms among VSA patients with intracoronary thrombi; however, logistic regression analyses did not clarify the specific characteristics in such patients. On the other hand, in the lesional analyses, intracoronary thrombi were detected only in the spastic segment. We also demonstrated that the atherosclerotic change and segmental spasm might account for thrombogenesis in the spastic segments.

Coronary spasm plays an important role in the occurrence of MINOCA [6,7,8,9] and much attention has focused on the relationship between intracoronary thrombi and coronary spasm [23,24,25]. There have been several possible mechanisms of the pathogenesis of intracoronary thrombi in VSA patients. Coronary plaque rupture, mediated by the coronary spasm’s external force, could be one such mechanism. This relationship was confirmed in autopsy case reports [26,27]. However, such plaque rupture has not been recognised in this and other studies [10,11,12,13,14]; thus, if present, this mechanism may be rare. Second, severe and repeating coronary spasms may lead to the repeating hypercontraction and extension of the intima, which results in coronary erosion. Using OCT, Tanaka et al. [28] demonstrated initial gathering (characterised by folding or gathering of the intima) at spastic segments during a spasm, which was the supportive finding responsible for the mechanism of intracoronary thrombus formation in the spastic segment. Finally, coronary spasms themselves may promote thrombus formation, mediated by the increased platelet aggregation in the coronary circulation [29] and/or an activated blood coagulation [30]. These factors, in a complex way, may contribute to thrombus formation at the spastic segments.

Using intracoronary modalities such as IVUS [11], OCT [12,13,14,31] and CAS [10,14], the frequency of intracoronary thrombus formation might differ, ranging from 0% to 40%. The kinds of adopted intracoronary modalities and/or the differences in the studied patients’ characteristics may contribute to the varied frequency of intracoronary thrombus formation. If patients with VA, a higher activity of VSA [22,32], were more frequently involved in the study, the frequency of intracoronary thrombus formation may have been higher, as shown in the study by Etsuda et al. [10]. In the present study, under the condition that the spastic segments could be assessed using CAS, the frequency of intracoronary thrombus formation was 30% per patient and 26% per coronary artery, keeping with the results reported by others [10,11,12,13,14]. In the present study, the frequency of VA was not different; however, the frequency of severe concomitant symptoms was higher in patients with intracoronary thrombi, indicating the possibility that they occurred according to the severity of the coronary spasm. In addition, male gender and smoking were significantly associated with intracoronary thrombi on univariate analysis. However, there was no statistically significant difference in logistic regression analysis. In general, females with VSA tend to have diffuse spasms [33], and this finding could account for the gender difference we observed. Further studies will be needed to confirm this finding.

Kitano et al. reported that intracoronary thrombi were more frequently observed in the spastic segments with focal spasms [14]. Several studies using IVUS have reported a thicker intima and/or presence of atherosclerosis at the spastic segment with focal spasms [34,35,36], leading to poorer prognoses in patients with focal spasms than those with diffuse spasms [16]. In the present study, intracoronary thrombi were detected at all spastic segments but not at all non-spastic segments, irrespective of the presence of atherosclerotic change, which indicates that coronary spasm plays an important role in thrombogenicity. Next, regarding the relationship between intracoronary thrombi and the spastic segments, atherosclerotic change and segmental spasm were defined as multiple focal spasms and meant almost the same thing as focal spasm at spastic segments, were the significantly associated factors. Such focal spasm and atherosclerotic changes may cause turbulent blood flow and increased blood coagulation ability, leading to intracoronary thrombosis. On the other hand, the frequency and degree of yellow plaque formation assessed using CAS at the spastic segment did not differ significantly at the spastic segment with intracoronary thrombi. The difference in the frequency of having lipid disorder and/or taking statins in the studied population may have contributed to this finding. Furthermore, a higher frequency of total or subtotal occlusion due to coronary spasm, and the tendency of provoked coronary spasm at low or moderate ACh doses, were recognised at the spastic segments with intracoronary thrombi, indicating that there was more coronary spasm at such lesions.

In the present study, VSA patients with intracoronary thrombi had a significantly higher frequency of severe concomitant symptoms associated with coronary spasms. In addition, the lesions with intracoronary thrombi were those with significantly more frequent focal spasms and total/subtotal spasms. These findings, including the patients’ characteristics and lesional characteristics, showed more coronary spasms in VSA patients with intracoronary thrombi. Thus, sufficient doses and kinds of coronary vasodilators should be administered in such patients. Conventionally, the effect of low-dose of aspirin (LDA) on the prognosis of VSA patients has not been fully clarified; however, several studies have shown no favourable effect of LDA in VSA patients without significant coronary stenosis [37,38]. The administration of LDA may not always be effective in all VSA patients and may be effective in a limited number of VSA patients. Judging from this study’s results, LDA for patients predisposed to intracoronary thrombi, such as those with severe concomitant symptoms, atherosclerotic changes and focal spasm, may be useful. Further studies will be needed to confirm this assertion.

There were several limitations to the present study. First, the study was performed on a small number of VSA patients and at only one institution. Second, we had no control group whose participants had negative SPT and had undergone CAS. Using OCT, Shin et al. demonstrated the presence of intracoronary thrombi in patients with negative SPT [13]. Thus, we could not compare thrombogenicity between patients with and without VSA. Third, the intracoronary observation using CAS in the present study was performed in the well-dilated coronary artery after an intracoronary administration of NTG. Thus, we could not observe the whole intracoronary wall sufficiently due to insufficient elimination of the coronary blood flow. Furthermore, as mentioned in the ‘Methods’ section, we sometimes experienced difficulties distinguishing between the intracoronary thrombus and the not-eliminated blood flow. Fourth, the co-registered system between CAS and fluoroscopic images could easily understand the position in a coronary artery; however, we could not understand the relationship between the spastic segment and the observation segment using CAS accurately. Fifth, we could not assess the frequency of intracoronary thrombus formation in VSA patients whose coronary spasms occurred in the distal segment of the coronary artery. We often have experienced such VSA patients with distal coronary spasms in the clinical setting. Due to the exclusion of such patients from the present study, our results could not show the frequency of intracoronary thrombus formation in the entire population of VSA patients. Sixth, we have focused on intracoronary thrombus formation as the pathogenesis of MINOCA; however, patients with MINOCA were not included in the present study. Thus, these intracoronary thrombi may be risk factors for INOCA, affecting refractory chest symptoms or microvascular angina. This hypothesis will be further investigated in further studies. Finally, we could not follow our studied patients sufficiently, neither could we objectively show their chest symptoms and/or clinical outcomes after their discharge from the hospital.

## 5. Conclusions

In the present study, we demonstrated that the rate of detection of intracoronary thrombi was 30% per patient and 26% per vessel in VSA patients whose spastic segments could be observed using CAS. More attention should be paid to possible intracoronary thrombi in VSA patients who have severe concomitant symptoms, as well as the atherosclerotic changes and focal spasms at the spastic segments. Our findings may spotlight coronary spasms as a significant mechanism in the pathogenesis of MINOCA.

## Figures and Tables

**Figure 1 diagnostics-11-01632-f001:**
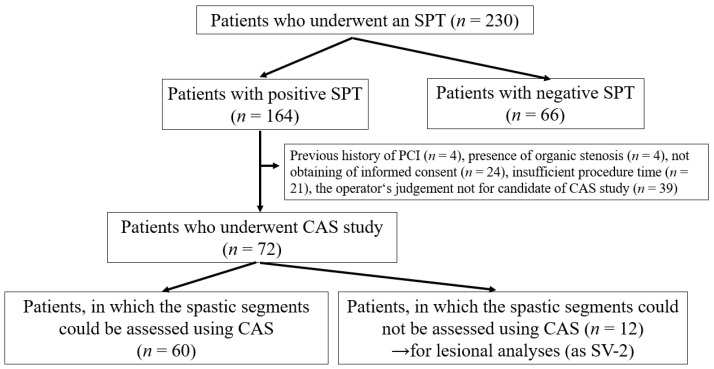
Study flowchart showing the retrospective enrolment of patients with positive SPT who underwent CAS. CAS, coronary angioscopy; SPT, spasm provocation test. We sometimes experienced difficulties discriminating the intracoronary thrombus and not-eliminated blood flow, as mentioned in the ‘Methods’ section.

**Figure 2 diagnostics-11-01632-f002:**
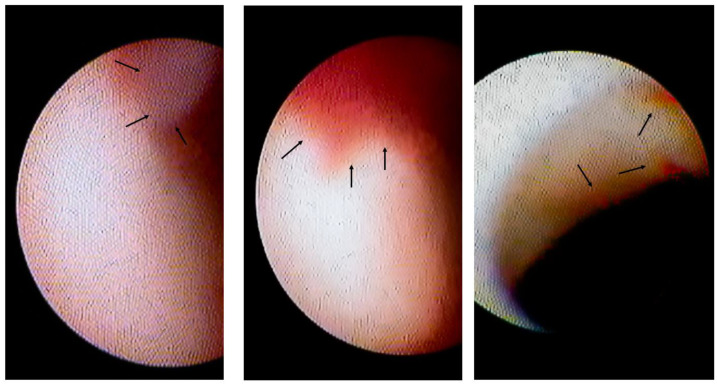
Intracoronary thrombi in VSA patients. The **left** panel shows the white thrombus (indicated by arrows) on the erosion-like red thrombus. The **centre** panel shows the erosion-like red thrombus, indicated by arrows. The **right** panel shows several erosion-like red thrombi, indicated by arrows.

**Figure 3 diagnostics-11-01632-f003:**
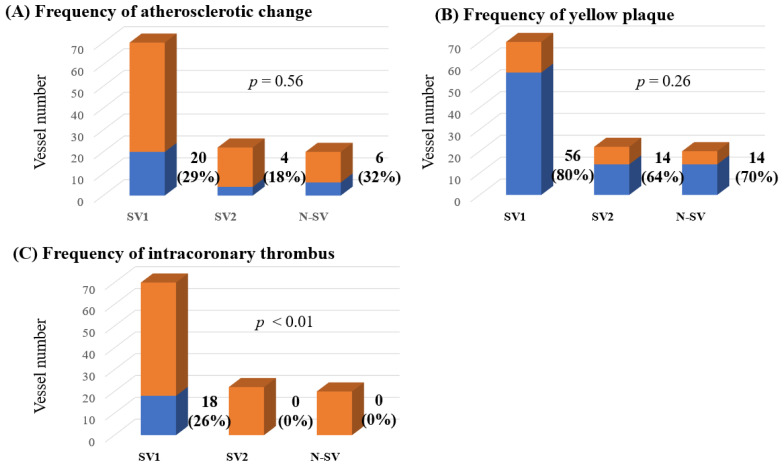
The frequencies of atherosclerotic change (**A**), yellow plaques (**B**) and intracoronary thrombi (**C**) among all lesions observed using CAS. CAS, coronary angioscopy; N-SV, non-spastic vessel; SV, spastic vessel. Among the SVs, those with a visible spastic segment using CAS were defined as SV-1, while those in which the spastic segment was invisible using CAS were defined as SV-2. The frequencies of atherosclerotic change and yellow plaques did not differ significantly between the three groups. However, intracoronary thrombi were detected only in the SV-1 (*p* < 0.01).

**Table 1 diagnostics-11-01632-t001:** Patients’ characteristics.

	Group I	Group II	
	Intracoronary Thrombus (−)	Intracoronary Thrombus (+)	*p*-Value
No. (%)	42 (70)	18 (30)	
Age (years)	65 ± 12	63 ± 10	0.6
Male/Female	16/26	14/4	<0.01
Body mass index	24.4 ± 4.4	24.8 ± 3.8	0.75
Coronary risk factors (%)			
Current smoker (%)	15 (36)	13 (72)	<0.01
Hypertension (%)	17 (40)	5 (28)	0.35
Dyslipidemia (%)	20 (48)	7 (39)	0.53
Diabetes mellitus (%)	6 (14)	14 (78)	0.45
Family history of CAD (%)	13 (31)	5 (28)	0.81
Chronic kidney disease (%)	9 (21)	4 (22)	0.95
LVEF (%) on echocardiogram	65 ± 8	65 ± 7	0.98

CAD, coronary artery disease; LVEF, left ventricular ejection fraction; No., number.

**Table 2 diagnostics-11-01632-t002:** Blood chemical parameters.

	Group I	Group II	*p*-Value
Total cholesterol (mg/dL)	210 ± 34	198 ± 39	0.24
Triglyceride (mg/dL)	120 ± 50	145 ± 79	0.14
High-density lipoprotein cholesterol (mg/dL)	64 ± 14	58 ± 19	0.24
Low-density lipoprotein cholesterol (mg/dL)	120 ± 34	107 ± 40	0.2
Fasting blood sugar (mg/dL)	105 ± 22	119 ± 45	0.1
Haemoglobin A1C (%)	5.9 ± 0.6	6.2 ± 1.0	0.23
eGFR (mL/min/1.73 m^2^)	70.7 ± 13.6	73.3 ± 19.9	0.56
D-dimer (µg/mL)	0.5 (0.5, 0.6)	0.5 (0.5, 0.5)	0.1
C-reactive protein (mg/dL)	0.07 (0.04, 0.20)	0.12 (0.03, 0.20)	0.5
Brain natriuretic peptide (pg/mL)	20 (10, 33)	17 (13, 40)	0.84

eGFR, estimated glomerular filtration rate.

**Table 3 diagnostics-11-01632-t003:** VSA-related parameters.

	Group I	Group II	*p*-Value
Medications before admission			
Coronary vasodilators (%)	16 (38)	4 (22)	0.23
Statins (%)	15 (30)	7 (39)	0.82
Aspirin (%)	6 (14)	3 (17)	0.81
Frequency of chest symptoms (/month)	4 (1, 2)	9 (2, 17)	0.36
Duration from onset to admission (month)	18 (4, 39)	15 (2, 84)	0.36
Variant angina (%)	0 (0)	1 (6)	0.12
Severely concomitant symptoms (%)	5 (12)	9 (50)	<0.01
Presence of multi-vessel spasm (*n*, %)	23 (39, 59)	9 (12, 75)	0.32

VSA, vasospastic angina.

**Table 4 diagnostics-11-01632-t004:** Frequencies of atherosclerosis, yellow plaque and intracoronary thrombus in all vessels.

	SV-1	N-SV	*p*-Value	SV-2	*p*-Value
			(SV-1 vs. N-SV)		(SV-1 vs. SV-2 vs. N-SV)
No.	70	20		22	
LAD/LCX/RCA	51/5/14	4/3/13	<0.01	12/0/10	<0.01
Atherosclerotic change on coronary angiogram (%)	20 (29)	6 (32)	0.8	4 (18)	0.56
Yellow plaque (%)	56 (80)	14 (70)	0.34	14 (64)	0.26
Degree of yellow plaque	1 (1, 2)	1 (0, 1)	0.2	1 (1, 1)	0.81
Intracoronary thrombus (%)	18 (26)	0 (0)	<0.01	0 (0)	<0.01

LAD, left anterior descending coronary artery; LCX, left circumflex coronary artery; No., number; N-SV, non-spastic vessel; RCA, right coronary artery; SV, spastic vessel. Among the SVs, where the spastic segment was visible using CAS were defined as SV-1, while those where the spastic segment was invisible using CAS were defined as SV-2.

**Table 5 diagnostics-11-01632-t005:** Frequencies of atherosclerosis, yellow plaque and intracoronary thrombi in 70 spastic vessels.

	Intracoronary Thrombus (−)	Intracoronary Thrombus (+)	*p*-Value
No.	52	18	
LAD/LCX/RCA	38/3/ 11	13/2/3	0.72
Atherosclerotic change on coronary angiogram (%)	9 (17)	11 (61)	<0.01
Yellow plaque (%)	41 (71)	15 (83)	0.68
Degree of yellow plaque	1 (1, 2)	1 (1, 2)	0.4
Focal spasm (%)	14 (27)	11 (71)	<0.01
Segmental spasm (%)	23 (44)	16 (90)	<0.01
Total or subtotal spasm (%)	13 (25)	9 (50)	0.04
Spasm-induced dose of ACh-L, M, H	6/32/14	6/10/ 2	0.07
ST-T elevation on ECG (%)	7 (13)	6 (33)	0.08

ACh, acetylcholine; ECG, electrogram; LAD, left anterior descending coronary artery; LCX, left circumflex coronary artery; No., number; RCA, right coronary artery. Low, moderate and high doses of ACh (L-ACh, M-ACh, H-ACh), were defined as 20, 50 and 80 μg for the RCA and 50, 100 and 200 μg for the LCA, respectively.

## Data Availability

Not applicable.

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
