# Peer review of "Intracoronary Thrombogenicity in Patients with Vasospastic Angina: An Observation Using Coronary Angioscopy"

_diagnostics, 2021, doi:10.3390/diagnostics11091632_

Round 1

Reviewer 1 Report

Teragawa et al. proposed an alternative and less expensive tool to diagnose intracoronary thrombi in patients with positive spasm provocation test, who underwent coronary angioscopy.

MINOR:

-P1, L28: casual instead of causal;

-P1, L36: Revision of the sentence "they have been reportedly";

-P8, L219: Better and exhaustive description of caption figure 3, table 4 and 5;

-P9, L241-242: Add reference;

-P9, L265-266: Revision of this sentence

MAJOR:

-P10, L284-287: Better and exhaustive description of the hypothesis;

-P10, L290: Better and exhaustive description of the results mentioned;

-Figure 1 (flow chart): Specify better this statement "Furthermore, we sometimes experienced difficulties discriminating the intracoronary thrombus and not-eliminated blood flow, as mentioned in the ‘Methods’ section"

Author Response

To reviewer number 1

Thank you very much for your valuable comments. Based on your comments, we have revised our manuscript as follows:

Teragawa et al. proposed an alternative and less expensive tool to diagnose intracoronary thrombi in patients with positive spasm provocation tests who underwent coronary angioscopy.

MINOR:

-P1, L28: casual instead of causal;

As per your comment, we have changed it to ‘causal’.

-P1, L36: Revision of the sentence "they have been reportedly";

Based on your suggestion, we have revised the sentence shown below.

Many investigations using intracoronary imaging techniques have been conducted

-P8, L219: Better and exhaustive description of caption figure 3, table 4 and 5;

Based on your comment, we have added some explanations to the captions of Figure 3, Table 4 and Table 5.

-P9, L241-242: Add reference;

As per your comment, we have added some references of case reports that showed the possibility of plaque rupture provoked by coronary spasm.

-P9, L265-266: Revision of this sentence

As per your comment, we have revised the sentence.

MAJOR:

-P10, L284-287: Better and exhaustive description of the hypothesis;

Based on your comment, we have added some explanations to this section.

In the present study, the proportion of VSA patients with intracoronary thrombi with severe concomitant symptoms related to coronary spasm was significantly higher than that of patients who did not. In addition, the lesions with intracoronary thrombi were more frequently associated with focal spasms and total/subtotal spasms. VSA patients with focal spasms had poorer prognoses than those with diffuse spasms. These findings, including the patients’ characteristics and lesional characteristics, showed more coronary spasms in VSA patients with intracoronary thrombi. Thus, sufficient doses and kinds of coronary vasodilators should be administered in such patients.

-P10, L290: Better and exhaustive description of the results mentioned;

As per your comment, we have added some explanations to this section.

The administration of LDA might not always be effective in all VSA patients. Judging from the present study’s results, LDA for the selected patients who had risk factors for intracoronary thrombi, such as severe concomitant symptoms, atherosclerotic changes and focal spasms could be useful. More studies are needed to further investigate this.

-Figure 1 (flow chart): Specify better this statement “Furthermore, we sometimes experienced difficulties discriminating the intracoronary thrombus and not-eliminated blood flow, as mentioned in the ‘Methods’ section”

According to your comments, we have added this sentence in the figure legend of Figure 1.

Reviewer 2 Report

  1. Line 55 – patient who had any chest symptoms to evauate their chest symptoms?? Please re write this sentence
  2. Please classify these 230 patients accordingly their presentation. Ie -how many had vasospastic angina as per definition or MINOCA as per definition?
  3. Present the results of spasm testing ie – frequency of LAD spasm, frequency of subtotal occlusion etc
  4. Line 61 – the excluded number of patients should be 92
  5. Line 172 – Please include the Group 1 details
  6. Please postulate why women are less likely to show evidence of thrombi.
  7. Given that this is a retrospective analysis, a prospective study adequately powered to detect statistical significance between groups are importanct. Could you comment on the number of patients requered to detect a statistical significance between groups/
  8. Please comment on the safety outcome of these procedures? Any adverse event recorded in these patients post spasm testing and angiooscopy?

Author Response

To reviewer number 2,

 Thank you very much for your valuable comments. As per your comments, we have revised our manuscript as follows:

1. Line 55 – patient who had any chest symptoms to evauate their chest symptoms?? Please re write this sentence

Based on your comments, we have deleted ‘to evaluate their chest symptoms’.

2. Please classify these 230 patients accordingly their presentation. Ie -how many had vasospastic angina as per definition or MINOCA as per definition?

Thank you for your comment. As shown in Figure 1, out of 230 patients, there were 164 patients with positive spasm provocation tests (SPT), and these 164 patients were predisposed to vasospastic angina (VSA). As you pointed out, there were four patients with myocardial infarction with the non-obstructive coronary artery (MINOCA); however, we did not perform a CAS study in MINOCA patients to avoid procedure-related complications. Regarding the latter findings, we have added these comments to the ‘Methods’ section.

3. Present the results of spasm testing ie – frequency of LAD spasm, frequency of subtotal occlusion etc

Thank you for your comments. We focused on the 112 provoked coronary arteries. The frequencies of LAD spasm and total/subtotal spasm were 63 (56%) and 23 (22 vessels in SV-1 and 1 vessel in SV; total: 21%), respectively. Based on these points, we have presented these findings in tables 4 and 5.

4. Line 61 – the excluded number of patients should be 92

Thank you for your comments. We have changed this number from 82 to 92.

5. Line 172 – Please include the Group 1 details

Based on your comments, we have added the following sentence.

There were 42 patients without intracoronary thrombi on CAS in Group I.

6. Please postulate why women are less likely to show evidence of thrombi.

Based on your comments, we have added the following sentences in the ‘Discussion’ section.

Male gender and current smoking were significantly associated with intracoronary thrombi on univariate analysis. However, there was no statistically significant difference on logistic regression analysis. In general, females with VSA tend to have diffuse spasms, and this finding could account for the gender difference we observed. Further studies will be needed to confirm this finding.

7. Given that this is a retrospective analysis, a prospective study adequately powered to detect statistical significance between groups are importanct. Could you comment on the number of patients requered to detect a statistical significance between groups/

Thank you for your valuable comments. We appreciate your opinion. However, we excluded some patients who could not provide informed consent or those whose coronary arteries were not suitable for a CAS study. Thus, we apologise for not calculating the required number of study participants for a prospective study.

8. Please comment on the safety outcome of these procedures? Any adverse event recorded in these patients post spasm testing and angiooscopy?

Thank you for your question. We recorded one patient whose coronary spasm recurred during a CAS study at the myocardial bridge segment. We have included this finding in the ‘Results’ section.

Reviewer 3 Report

In a study by Teragawa et al, the authors aimed to assess the phenomenon of intracoronary thrombi in patients with vasospastic angina with the use of coronary angioscopy. They enrolled 60 patients with VSA in which intracoronary thrombi on CAS were found. Intracoronary thrombi were detected in 30% of VSA patients.

I have following remarks:

  1. In introduction authors should discuss more on INOCA than MINOCA in terms of VSA, including results from the Cormica study.
  2. In line 42 authors stated that IVUS and OCT could help determine the frequency of intracoronary thrombus formation in VSA patients. In 40 MHz IVUS it’s not possible to confirm thrombus formation only to have strong suspicion esp after postprocessing with gain play. HD-IVUS (60 MHz) is way better. OCT is the best tool for that purpose.
  3. Why diagnosis of VSA is so important? It’s not only due to medication adjustment but also higher risk of sudden cardiac death.
  4. Please specify whether only pts with chronic coronary syndromes were enrolled or pts with ACS or mix?
  5. Was ischemia confirmed before coronary angio (SPECT?)
  6. 71% pts with VSA is an extremely high percent. Please comment on that.
  7. Was a 12 lead ECG monitoring engaged during SPT?
  8. I suppose ic bolus of Ach was administered a not an infusion. Please correct the methods section.
  9. Which software was used for QCA assessment or was it just visual assessment?
  10. Did the authors collect any data with angina/QoL questionnaires after treatment modification?
  11. Do the authors have some data on clinical outomes? 30d/ 6 months/1 year? This would be really interesting.
  12. In discussion section authors focus again on MINOCA and not INOCA, two different clinical scenarios.
  13. Major English language is required.

Author Response

To reviewer number 3,

Thank you very much for your valuable comments. Based on your comments, we have revised our manuscript as follows:

In a study by Teragawa et al., the authors aimed to assess the phenomenon of intracoronary thrombi in patients with vasospastic angina using coronary angioscopy. They enrolled 60 patients with VSA who had intracoronary thrombi on CAS. Intracoronary thrombi were detected in 30% of VSA patients.

I have the following remarks:

1. In introduction authors should discuss more on INOCA than MINOCA in terms of VSA, including results from the Cormica study.

Based on your insightful comments, we have added INOCA into the ‘Introduction’ section. However, in the present study, we focused on intracoronary thrombi in VSA patients, and we focused more on MINOCA rather than on INOCA. Nevertheless, if you think it would be better to focus on INOCA, please do not hesitate to let us know to make the necessary changes.

2. In line 42 authors stated that IVUS and OCT could help determine the frequency of intracoronary thrombus formation in VSA patients. In 40 MHz IVUS it’s not possible to confirm thrombus formation only to have strong suspicion esp after postprocessing with gain play. HD-IVUS (60 MHz) is way better. OCT is the best tool for that purpose.

According to your comments, the methods shown above could be more effective in detecting intracoronary thrombi. As of 2021, we have adopted OCT and have not used HD-IVUS at our institution. However, regarding CAS, we have been familiar with this methodology since 2014, and we carried out the present study retrospectively during this period. However, we have mentioned HD-IVUS in the ‘Discussion’ section.

3. Why diagnosis of VSA is so important? It’s not only due to medication adjustment but also higher risk of sudden cardiac death.

Thank you for your valuable comments. We agree with your statement and have added the following sentence to the ‘Methods’ section.

The aim of performing SPT was to clarify the origin of chest symptoms to provide appropriate medication, which would lead to the prevention of sudden cardiac death.

4. Please specify whether only pts with chronic coronary syndromes were enrolled or pts with ACS or mix?

Thank you for your valuable comments. We have added the comments regarding MINOCA that were excluded from the present CAS study.

5. Was ischemia confirmed before coronary angio (SPECT?)

Thank you very much for your insightful comment. We had some information about myocardial scintigraphy in some of the studied VSA patients. However, the kinds of myocardial scintigraphy and the time lapses between myocardial scintigraphy and coronary angiography varied a lot; therefore, we did not have sufficient data to report.

6. 71% pts with VSA is an extremely high percent. Please comment on that.

Thank you very much for your comment. As much as possible, we tried to identify patients with symptoms in the outpatient clinic and send those who were likely to have VSA to undergo the spasm provocation test (SPT). Although it was not included in the text, the positive rate of SPT for the past nine years was 68% (382/561).

7. Was a 12 lead ECG monitoring engaged during SPT?

Thank you for your question. We had ECG monitoring during the spasm provocation test. Regarding the ST-T segment elevation on ECG during coronary spasm, we have added this information in the ‘Results’ section and Table 5.

The frequency of ST-T segment elevation on ECG during coronary spasm tended to be higher in the spastic segments with intracoronary thrombi than in those without intracoronary thrombi (P = 0.08).

8. I suppose ic bolus of Ach was administered a not an infusion. Please correct the methods section.

Based on your comments, we have changed the term from ‘infusion’ to ‘administration’.

9. Which software was used for QCA assessment or was it just visual assessment?

Based on your comments, we have added the following sentences into the ‘Methods’ section.

We selected spastic and atherosclerotic segments for quantitative analysis. In all cases, the luminal diameters were measured by a single investigator blinded to the clinical data, using an end-diastolic frame in a computer-assisted coronary angiographic analysis system (CAAS II/QUANTCOR; Siemens, Berlin, Germany).

10. Did the authors collect any data with angina/QoL questionnaires after treatment modification?

Thank you for your question. We admit it was important to collect such data. Unfortunately, we have no information about angina/QoL and the patients’ ensuing clinical outcomes (No.11). However, we have included this point in the ‘Study limitations’ section.

11. Do the authors have some data on clinical outomes? 30d/ 6 months/1 year? This would be really interesting.

Thank you for your question. Unfortunately, we have no such available data. However, we have included this point in the ‘Study limitations’ section.

12. In discussion section authors focus again on MINOCA and not INOCA, two different clinical scenarios.

Thank you very much for your insightful comment. We acknowledge the importance of the point you are raising. However, focusing on intracoronary thrombi as one of the pathogenetic mechanisms of MINOCA enabled us to discuss this problem mainly in the ‘Discussion’ section. Thus, we have mentioned this problem associated with the relationship between intracoronary thrombi and INOCA, in the ‘Study limitations’ sections as presented below. However, if you are of the opinion that we should discuss this issue in more depth, please let us know so that we can discuss it more.

Furthermore, we have focused on intracoronary thrombus formation as the pathogenetic mechanism of MINOCA. However, patients with MINOCA were not included in the present study. Thus, this intracoronary thrombus may be the factors of INOCA, affecting refractory chest symptoms or microvascular angina. These will be further investigated in future studies.

13. Major English language is required.

Thank you very much for your comment. We have solicited English Language editing services from ENAGO. The manuscript will be reviewed by a team of editors who are native English speakers.

Round 2

Reviewer 1 Report

This revised manuscript is well structured and presente.

Reviewer 3 Report

Thank you for providing revision. I have no further remarks.